# Association between Breakfast Skipping and Body Weight—A Systematic Review and Meta-Analysis of Observational Longitudinal Studies

**DOI:** 10.3390/nu13010272

**Published:** 2021-01-19

**Authors:** Julia Wicherski, Sabrina Schlesinger, Florian Fischer

**Affiliations:** 1School of Public Health, Bielefeld University, 33615 Bielefeld, Germany; julia.wicherski@bfarm.de; 2German Diabetes Center, Institute for Biometry and Epidemiology, Leibniz Center for Diabetes Research at Heinrich Heine University, 40225 Düsseldorf, Germany; sabrina.schlesinger@ddz.de; 3Institute of Public Health, Charité–Universitätsmedizin Berlin, 10117 Berlin, Germany; 4Institute of Gerontological Health Services and Nursing Research, Ravensburg-Weingarten University of Applied Sciences, 88250 Weingarten, Germany

**Keywords:** breakfast skipping, overweight, obesity, weight gain, Body Mass Index (BMI) change, systematic review, meta-analysis, observational longitudinal studies

## Abstract

Globally, increasing rates of obesity are one of the most important health issues. The association between breakfast skipping and body weight is contradictory between cross-sectional and interventional studies. This systematic review and meta-analysis aims to summarize this association based on observational longitudinal studies. We included prospective studies on breakfast skipping and overweight/obesity or weight change in adults. The literature was searched until September 2020 in PubMed and Web of Science. Summary risk ratios (RRs) or β coefficients with a 95% confidence interval (CI), respectively, were estimated in pairwise meta-analyses by applying a random-effects model. In total, nine studies were included in the systematic review and three of them were included in the meta-analyses. The meta-analyses indicated an 11% increased RR for overweight/obesity when breakfast was skipped on ≥3 days per week compared to ≤2 days per week (95% CI: 1.04, 1.19, *n* = two studies). The meta-analysis on body mass index (BMI) change displayed no difference between breakfast skipping and eating (β = −0.02; 95% CI: −0.05, 0.01; *n* = two studies). This study provides minimal evidence that breakfast skipping might lead to weight gain and the onset of overweight and obesity.

## 1. Introduction

Nowadays, the world is experiencing an obesity epidemic. In 2016, 39% of adults were overweight and 13% were obese, worldwide. Thus, obesity’s prevalence is three-fold higher than in 1975 [1] and it is still rising. For example, the prognosis for the United Kingdom (UK) estimates that approximately 60% of men and 50% of women will be obese by 2050 [2]. Today, the majority of countries around the world are affected by obesity prevalence rates above 10% and estimates suggest a rise to 20% of world’s population being affected by obesity by 2025 [3,4]. Globally, this rapid increase in the prevalence of overweight and obesity is one of the most important health issues [5,6].

Obesity is a major contributor to the global burden of disease through its deuteropathies of serious non-communicable diseases (NCDs) [7]. Psychological, pulmonary, orthopaedical, cardiovascular, metabolic, reproductive, and oncological diseases are attributable to obesity. For example, obesity is associated with depression, obstructive sleep apnoea, osteoarthritis, myocardial infarction, diabetes mellitus type 2, infertility, and colon cancer [8,9,10,11,12,13]. Therefore, obesity may cause premature death. In 2015, obesity contributed to 4 million deaths, equivalent to 7.1% of all-cause mortality, worldwide [7]. Furthermore, there is a huge economic and social burden of obesity. Total health costs and drug costs increase with increasing body mass, which is proportionally beyond their standard values. Obesity correlates to a low socioeconomic status, as well [14,15,16,17,18,19,20]. Therefore, one target of the World Health Organization’s (WHO) Global NCD Action Plan 2013–2020 is to halt the rise in obesity by 2025 [21]. Moreover, actions against obesity are necessary to achieve the third Sustainable Development Goal which comprises the target to decrease the number of premature deaths caused by NCDs by 33.3% by 2030 [22].

The underlying pathological process of obesity is represented by the increase in both the total number and size of fat cells, which leads to a heightened accumulation of fat cells in relation to one’s body size [8,23]. Being overweight is defined as elevated body fat accumulation, while obesity defines a situation characterized by an excess body fat mass [1,24,25]. The most common used measurements to assess human body size are anthropometric measures (e.g., body mass index (BMI) or waist–hip ratio (WHR)) and measurements of body composition (e.g., bioelectrical impedance analysis (BIA) or dual-energy x-ray absorptiometry (DEXA)). However, there are many more methods available for assessing human body weight status [26,27,28,29,30].

The etiology of obesity is multifactorial, but the fundamental determinant is the positive energy balance. This is mostly determined by a high energy intake through inappropriate nutrition and a low energy expenditure through physical inactivity [24,25]. In view of nutritional physiology, it is notable that breakfast is the meal eaten after the longest period with an empty stomach (i.e., postprandial fasting), and therefore, it has the potential to decrease the risk of weight gain due to several metabolic mechanisms [31,32,33]. For example, reduced levels of ghrelin (growth hormone release inducing, an appetite suppressant peptide) and increased postprandial energy expenditure have been observed when breakfast is eaten. Moreover, a hypothesis exists stating that nutrient timing is part of the circadian rhythm. In the scope of breakfast skipping, negative effects on the circadian rhythm—such as the irregulating of metabolism—with consequences related to weight management, are conceivable [32]. Additionally, international recommendations on breakfast agree in their statements that daily consumption of breakfast is advisable for providing a sufficient intake of macro- and micronutrients, maintaining body weight, and improving cognitive functions [34,35,36,37,38]. In contrast, breakfast skipping is associated with elevated plasma lipoproteins and fasting glucose [19,38], and insufficient intake of micronutrients [39].

Considering the huge medical, financial, and social burden of obesity, this study aims to examine whether breakfast skipping is associated with adult body weight. Existing systematic reviews and meta-analyses on this topic have examined target groups including children and adolescents in all study designs [40,41,42,43,44,45,46,47,48,49,50,51,52]. However, adults have only been studied in cross-sectional [53,54,55] and interventional study designs [56,57,58,59,60,61,62,63]. Interventional studies comprise the highest level of evidence but are limited to comparably young and healthy participants, analyzed in small sample sizes under laboratory conditions. For this reason, the systematic review and meta-analysis presented here is based on primary studies using observational longitudinal study designs, to gain further evidence with a high level of external validity.

## 2. Materials and Methods

A systematic review and subsequent meta-analyses were conducted. The study was planned and conducted in accordance with the “Meta-analysis Of Observational Studies in Epidemiology” (MOOSE) standards [64]. The systematic literature search, screening of the identified literature, data extraction and quality assessments were carried out independently by two reviewers (Julia Wicherski, Florian Fischer). There were no discrepancies in judgement between the two independent reviewers.

### 2.1. Search Strategy

According to the “Population-Item-Comparison-Outcome” (PICO) framework, the population of interest was exclusively adults—from all around the world—aged 18 years or older. The exposure of interest was breakfast skipping. This was compared to breakfast eating as regards the occurrence of overweight, obesity or weight gain, respectively. The literature review was conducted in PubMed and Web of Science and included all literature related to the topic that was published up until September 2020.

The included studies were from observational longitudinal study designs and had specified effect estimates expressed as risk ratios (RRs), such as odds ratios, hazard ratios, or relative risk. These RRs were reported with corresponding 95% confidence intervals (95% CI). Alternatively—instead of RRs—some of the included studies reported a coefficient with corresponding 95% CI, in the case that the regression models contained a linear term, such as continuous variables for breakfast frequency and/or BMI. Participants of the included studies were aged ≥ 18 years. The outcome was measured through BMI, waist circumference (WC), WHR, waist-height ratio (WHtR), body fat percentage (BF%), or weight change (gain or loss). The included studies were published/peer-reviewed articles and written in English or German. Studies were excluded if the study was an interventional study, a cross-sectional study, a case study or case report. Likewise, studies were excluded if they did not have specified effect estimates expressed as RRs or coefficients. Individuals who were aged < 18 years (e.g., children, adolescents) or who were pregnant were also excluded.

To build our search queries, we used the Boolean operators “AND” and “OR”. Furthermore, we used truncations to search for all terms that begin with a word of interest (e.g., using obes* to find obesity as well). We searched in specific fields like “Mesh” or “Publication Type” or “Title (TI)” or “Topic (TS)”, and we used abbreviations of effect estimates in our search (i.e., odds ratio (OR), hazard ratio (HR) and relative risk (RR)). The following query was used for PubMed: “(((((((breakfast* OR “breakfast skipping” OR “breakfast frequency” OR “breakfast omission” OR “breakfast” (Mesh) OR ((breakfast* AND (skipp* OR frequen* OR omit* OR omis* OR consum*))) AND (“body weight” (Mesh) OR “overweight” (Mesh) OR “obesity” (Mesh) OR “body weights and measures” (Mesh) OR “body weight” OR “body fat” OR “body mass” OR *weight OR obes* OR adipos* OR “BMI” OR “WHR” OR “WC” OR “WHtR” OR waist OR circumference OR “body size” OR “body fat distribution” (Mesh)) AND (cohort* OR “cohort studies” (Mesh) OR “case control studies” (Mesh) OR “OR” OR “RR” OR “HR” OR retrospective OR prospective OR observational OR “longitudinal studies” (Mesh) OR “follow-up studies” (Mesh) OR “Observational Studies as Topic” (Mesh) OR “Observational Study” (Publication Type))))))))”.

For Web of Science, the following query was utilized: “((TI = (breakfast*) AND TI = (skipp* OR omi* OR freq* OR eat*) OR (TI = (“breakfast skipping” OR “breakfast omission” OR “breakfast frequency”) OR TS = (“breakfast skipping” OR “breakfast omission” OR “breakfast frequency”))) AND (TI = (*weight OR obes* OR adipos* OR fat OR mass OR “body weight” OR “body fat” OR “body mass”) OR TS = (*weight OR obes* OR adipos* OR fat OR mass OR “body weight” OR “body fat” OR “body mass”)))”.

### 2.2. Data Extraction and Quality Assessment of Included Studies

Information on the first author’s surname, year of publication, country, study name, design and follow-up period, total number of participants, number of cases, distribution of sex and mean age, exposure and outcome definitions and measurements, specified effect estimates—expressed as RRs or coefficients—with the corresponding 95% CI, and adjusted covariables were extracted from each study and included in the qualitative and quantitative syntheses.

The risk of bias in included studies was assessed by applying the “Risk Of Bias In Non-randomized Studies of Interventions” (ROBINS-I) tool [65] for each study (Appendix A). Along with the risk of bias due to confounding, age, sex, education or socioeconomic status, smoking, alcohol, physical activity and total energy intake (TEI) were established in accordance with the literature [31,34,40,66,67,68,69,70,71,72,73,74,75,76,77,78,79,80,81]; as these are important covariables because of their confounding nature of being associated with breakfast and body weight. Moreover, the quality of evidence of the conducted meta-analyses was assessed by using the “Nutrition Grading of Recommendations Assessment, Development and Evaluation” (NutriGRADE) approach [82] (Appendix A).

### 2.3. Statistical Analysis

Pairwise meta-analyses were conducted by comparing skipping breakfast on ≥3 days per week to ≤2 days per week, and skipping breakfast to eating breakfast without detailed category definition, respectively. The first meta-analysis for breakfast skipping focused on the occurrence of overweight (defined as BMI ≥ 25 kg/m^2^) and/or obesity (defined as BMI ≥ 30 kg/m^2^). The second meta-analysis for breakfast eating versus skipping focused on changes in BMI (change in kg/m^2^).

Due to the fact that there are no standard definition criteria for breakfast skipping and eating in regard to its frequency, exposure definitions of this meta-analyses were adopted from the definitions used by the primary studies.

A random-effects model was applied due to the assumption that a normal distribution of the true effect and the heterogeneity within and between the studies was caused by unexpected effects rather than residual effects [83,84]. At first, the natural logarithm of each risk estimate expressed as a risk ratio (logRR) for overweight/obesity, respectively, was calculated. Subsequently, the logRR for each included study was weighted and pooled accordingly to the variance-based method of DerSimonian and Laird [85], which considers the variability within and between the studies. In the scope of the second meta-analysis, β regression coefficients were weighted and pooled directly.

Moreover, for each of the two meta-analyses conducted, we performed a sensitivity analysis for the standard error adjustment by applying the Knapp–Hartung method. This ensured consideration of the less favorable statistical properties of the DerSimonian and Liard method in meta-analyses with a small number of included studies [86] and the heterogeneity of included studies, as was the case in our analyses.

We used the I^2^ test to evaluate the heterogeneity. I^2^ is a measure of inconsistency which describes the variability between the studies included in the meta-analysis due to heterogeneity rather than chance [87]. According to the Cochrane recommendation [88], analyses should include ≥ 10 studies to check for publication bias. Since only two studies were included in the meta-analyses, funnel plots and Egger’s test were not applied. Checking for missing studies by applying the trim and fill analysis was also deemed to not be practicable. The meta-analyses were conducted by using the metan and metareg package in Stata v16.1 (College Station, TX, USA).

## 3. Results

### 3.1. Identified Literature and Characteristics of Included Studies

Overall, 3294 records were obtained through database screening and an additional 23 records were identified due to searching reference lists. In scope of the inclusion criteria, 2952 records were excluded. Out of the 180 remaining articles, 171 were excluded after full-text screening. Most excluded studies did not have an observational longitudinal study design (*n* = 68), or their exposure did not match with the corresponding inclusion criterion (*n* = 30). Further studies were excluded because they were published in languages other than English or German (*n* = 22), or because their outcome (*n* = 21) or both their exposure and outcome did not match with the corresponding inclusion criterion (*n* = 15). Furthermore, another nine studies were excluded due to participants being < 18 years old or pregnant, another four studies were excluded as they specified no effect estimates or other effect estimates than those that were required, and two studies were excluded because no full text was available and their authors could not be reached. Figure 1 exemplifies the literature screening process according to the Preferred Reporting Items for Systematic Reviews and Meta-Analyses (PRISMA) [89] flowchart.

Consequently, nine studies were included in the qualitative synthesis. As a result of this synthesis, four studies were not comparable in a meta-analysis due to a too great a difference in their exposure or outcome categories, respectively. Accordingly, Guinter et al. [90] used the breakfast eating category of 3–4 days per week as reference and compared this to 0, 1–2, 5–6, and 7 days per week. Due to this reference category, it was not practicable to pool this study result with the results of Hurst and Fukuda [37] and Kito et al. [38]. Odegaard et al. [91] analyzed BMI ≥ 30 kg/m^2^ and reported hazard ratios. Therefore, too much heterogeneity was present to pool this study result with the results obtained by Hurst and Fukuda [37] and Kito et al. [38], because they analyzed BMI ≥ 25 kg/m^2^ and reported odds ratios. In view of linear models, the studies by Nooyens et al. [92] and Smith et al. [39] were not comparable, because of their very different exposure categories: continuous breakfast frequency 0–7 days per week versus the dichotomous met/not met breakfast guideline. Lastly, three studies were included in the quantitative synthesis consisting of two meta-analyses (see Figure 1). In accordance with this screening process, Appendix A lists the studies that were excluded, sorted by exclusion criteria.

The characteristics of the nine studies included in the review are shown in Table 1. The subsequent Table 2 is an addition to Table 1 and displays the confounders adjusted in the included studies.

The studies were published between 2005 and 2020. They were conducted in Japan (*n* = 3), the USA (*n* = 4), Canada (*n* = 1), the Netherlands (*n* = 1) and Australia (*n* = 1). All nine studies had an observational longitudinal study design and their follow-up duration ranged from 5 years to 18 years. Out of the nine studies, three studies were retrospective cohort studies [36,37,38], while the other five studies were prospective cohort studies [39,90,91,92,93,94]. In four studies, only male participants were involved in the analyses [36,38,92,94]. Guinter et al. [90] only involved female participants. The remaining four studies included both genders, but proportionally, females were represented to a greater degree in three out of these four studies [39,91,93]. The mean age of the participants ranged from 21.5 years to 58 years. Odegaard et al. [91], Smith et al. [39] and Nooyens et al. [92] recruited citizens who lived within their analyzed regions, whereby the latter study only included male citizens. Kahleova et al. [93] recruited participants within Adventist church members, while van der Heijden et al. [94] enlisted male health professionals. Hurst and Fukuda [37] and Kito et al. [38] used health insurance data, whereby the latter study enlisted only male participants. Guinter et al. [90] recruited “healthy” sisters of women diagnosed with breast cancer (healthy in the way of not also having a cancer diagnosis).

The categorization of the exposure of breakfast skipping varied between the studies. Hurst and Fukuda [37], Kito et al. [38], and Odegaard et al. [91] defined breakfast as skipped when participants were skipping on ≥3 days per week, while Goto et al. [36] defined breakfast as skipped already when participants skipped breakfast on ≥2 days per week. Moreover, Kahleova et al. [93], Smith et al. [39] and van der Heijden et al. [94] defined breakfast as skipped or eaten without any specification of days per week. Nooyens et al. [92] used a continuous exposure variable defined as breakfast skipped on 0 days to 7 days per week, while Guinter et al. [90] used breakfast eating frequency categories, such as eating breakfast on 0, 1–2, 3–4, 5–6, and 7 days per week. Concerning the exposure measurement, only five studies used a validated tool [38,39,90,91,92] (see Table 2). With regard to the underlying definition of breakfast, most studies [36,37,38,90,92,94] reported no definition. The study by Odegaard et al. [91] reported that they did not have an explicit definition of breakfast. By contrast, Kahleova et al. [93] defined breakfast as “a meal eaten between 05:00 and 11:00”, whereas Smith et al. [39] defined breakfast skipping as “not eating a snack, small meal or large meal between 6:00 and 9:00 AM”.

In view of the outcome, six studies used BMI as measure for body weight [36,37,38,90,91,93]. Out of them, Hurst and Fukuda [37], Kito et al. [38], Odegaard et al. [91], and Guinter et al. [90] used a BMI cut-off to define the body weight of participants as overweight or obese, respectively. Accordingly, overweight was defined as a BMI ≥ 25 kg/m^2^ in Hurst and Fukuda [37], Kito et al. [38] and Guinter et al. [90]. Obesity was defined as a BMI ≥ 30 kg/m^2^ in Odegaard et al. [91] and Guinter et al. [90]. On the other hand, Goto et al. [36], Kahleova et al. [93] and Hurst and Fukuda [37] analyzed changes in BMI values over different periods. Nooyens et al. [92], Smith et al. [39], van der Heijden et al. [94] and Guinter et al. [90] investigated the outcome of weight change by analyzing changes in kilograms of body weight over different periods. Moreover, three studies also used the WC as an outcome measure for overweight or weight change [37,91,92]. Other measures of body weight (i.e., WHtR or WHR or BF%) were not utilized in the included studies. An exception was the study by Guinter et al. [90], in which they utilized the measures WC and WHR, but unfortunately this was only for cross-sectional outcomes, not for incident outcome measurement.

In view of the outcome measurement, in seven out of nine studies the outcome values were measured by trained stuff [36,38,39,90,91,92,93]. In three out of these six studies, self-reported information on weight and height were additionally collected [39,90,93]. Guinter et al. [90] measured only using trained staff at baseline and collected the self-reported weight at baseline and follow-up. In contrast, van der Heijden et al. [94] used only self-reported weight values. Within the study by Hurst and Fukuda [37] it was not clear whether BMI or WC values were measured or self-reported (see Table 1).

In addition to Table 1, Table 2 displays the adjusted confounders of the included studies. Only two studies [90,91] adjusted their results for all of the important confounders (see Table 2).

### 3.2. Association between Breakfast Skipping and Body Weight

Regarding the analyzed association between breakfast skipping and body weight, all nine studies showed an increased relative risk regarding overweight/obesity. Within this, seven out of nine studies reported that skipping breakfast was associated with weight gain [36,37,38,39,91,93,94]. Nooyens et al. [92] described that with increasing frequency of breakfast eating, weight gain also increased. Guinter et al. [90] described that eating breakfast rarely (i.e., 0 or 1–2 d/wk) and eating breakfast often (i.e., 5–6 or 7 d/wk) decreased the risk for a 5-year incident of weight gain, compared to having an inconsistent breakfast eating frequency (i.e., eating breakfast on 3–4 d/wk). Focusing on the seven studies with inverse associations, four studies described that eating breakfast decreased the relative risk of overweight, obesity, gain of weight, BMI, or WC, respectively [37,91,93,94]. The other three studies described the same inverse correlation, but the other way around. Breakfast skipping increased the relative risk of overweight, obesity, gain of weight, BMI, or WC, respectively [36,38,39] (see Table 1).

Within the scope of increased BMI values, Guinter et al. [90] reported a 1.35 times higher relative risk for the incident of a 5-year BMI ≥ 25 kg/m^2^ for women who ate breakfast on 3 to 4 days per week than women who ate breakfast on 0 days per week (i.e., 1/RR = 1/0.74 = 1.35). However, on the other hand, the same analysis displayed a 12% decreased risk for the incident of a 5-year BMI ≥ 25 kg/m^2^ for women who ate breakfast on 7 days per week, compared to women who ate breakfast on 3 to 4 days per week (i.e., RR = 0.88). Similar directions were observed for the 5-year incident of a BMI ≥ 30 kg/m^2^ within this analysis of The Sisters Study (see Table 1). By contrast, Odegaard et al. [91] reported a consistently increasing trend in the risk of becoming obese (i.e., BMI ≥ 30 kg/m^2^) for breakfast skipping: people who ate breakfast only on ≤3 days per week had a 1.33-fold higher risk of obesity than people who ate breakfast on 4–6 days per week (i.e., 1/HR = 1/0.75 = 1.33). The protective result as a consequence of breakfast eating behavior was even clearer when people who ate breakfast on 7 days per week were compared to people who ate breakfast only on ≤3 days per week: the latter showed a 75% increased risk of obesity (i.e., 1/HR = 1/0.57 = 1.75). Moreover, Goto et al. [36] reported that a >5% increase in BMI value was 1.34 times more likely in men who skipped breakfast on ≥2 days per week compared to men who skipped breakfast only on ≤1 day per week.

In view of weight change, The Sisters Study [90] displayed no association between breakfast eating frequency categories and the 5-year incident of ≥5 kg weight gain. Nooyens et al. [92] found no linear association between breakfast skipping and weight change, whereas Smith et al. [39] reported an increase in body weight when breakfast was skipped. Accordingly, people who did not meet the guidelines on breakfast (i.e., they skipped breakfast), gained in general 1.5 kg weight over a period of 5 years compared to people who met the guidelines on breakfast (i.e., they ate breakfast) [39]. Additionally, the study by van der Heijden et al. [94], men who skipped breakfast showed a 15% increased risk of ≥5 kg weight gain compared to men who did eat breakfast (i.e., 1/HR = 1/0.87 = 1.15).

As regards the measurements of abdominal obesity, WC was used in some of the included studies. The results of the Doetinchem Cohort Study displayed that the WC of Dutch men increased by about 0.10 cm for each day on which breakfast was skipped [92]. In the study by Hurst and Fukuda [37], there was no linear association between breakfast skipping and WC change. In contradiction to these two studies, the results of the Coronary Artery Risk Development in Young Adults (CARDIA) study showed that the hazard ratio for abdominal obesity was 0.84 (95% CI: 0.70; 0.99) when breakfast was consumed on 4 to 6 days per week, and 0.78 (95% CI: 0.66; 0.91) when breakfast was consumed on 7 days per week. Both were compared to participants who consumed breakfast on ≤3 days per week [91].

#### 3.2.1. Association between Breakfast Skipping and Overweight/Obesity

Two studies comprised the meta-analysis on breakfast skipping on ≥3 days compared to ≤2 days per week and the occurrence of overweight/obesity, involving 25,764 cases among 105,251 participants (Figure 2).

The pooled effect estimate expressed as a summary risk ratio (RR) for the occurrence of overweight/obesity is 1.11 (95% CI: 1.04, 1.19; I^2^ = 24.9%, *p* heterogeneity = 0.249; *n* = 2 studies). Accordingly, participants who skipped breakfast on ≥3 days per week have an increased relative risk of 11% for becoming overweight/obese compared to participants who skipped breakfast on ≤2 days per week. In view of the inconsistency of this meta-analysis, there is a 24.9% level of variability between the included studies due to heterogeneity.

As a result of this adjustment, the corresponding 95% CI widens, the estimation becomes less precise, and the identified association between breakfast skipping and overweight/obesity in the meta-analysis becomes lost (adjusted RR = 1.11; 95% CI: 0.72, 1.72).

#### 3.2.2. Association between Breakfast Skipping and BMI Change

The meta-analysis on the association between breakfast eating compared to breakfast skipping and BMI change comprised two studies, involving 108,413 participants (Figure 3).

The pooled β coefficient is −0.02 (95% CI: −0.05, 0.01; I^2^ = 58.1%, *p* heterogeneity = 0.123; *n* = 2 studies). It appears that breakfast eaters showed a slight decrease in BMI. However, the precision of this estimate is low and there is no association in BMI change between breakfast eating and skipping. In view of the inconsistency of this meta-analysis, there is a 58.1% level of variability between the included studies due to heterogeneity.

Moreover, the standard error adjustment in our sensitivity analysis results in an even wider 95% CI and less precise estimation (adjusted β = −0.02; 95% CI: −0.20, 0.16).

### 3.3. Quality of Included Studies

To assess the risk of bias in the included studies, ROBINS-I tool was applied for each study. In accordance, only two studies were judged as having a moderate risk of bias [90,91], another six studies had a serious risk of bias [36,38,39,92,93,94], and one study was assigned as having a critical risk of bias [37] (see Appendix A). To assess the quality of evidence of the two conducted meta-analyses, NutriGRADE was used. In accordance, both meta-analyses were assigned with a very low meta-evidence score. Appendix A provides an overview of the assessed domains and points achieved for each domain and meta-analysis.

## 4. Discussion

All nine studies included in the review reported a statistically significant association between breakfast skipping and overweight/obesity or weight gain. Moreover, eight out of nine studies displayed that breakfast skipping increased the relative risk for overweight/obesity or weight gain, respectively, compared to eating breakfast. Both meta-analyses provided a very low meta-evidence; one showed an increased relative risk for overweight/obesity, while the other might imply a small tendency for weight gain displayed as increasing BMI values, when breakfast is skipped. Skipping breakfast on ≥3 days per week increased the risk to become overweight/obese about 11% (95% CI: 4%, 19%) compared to skipping breakfast on ≤2 days per week. The second meta-analysis showed no association between breakfast skipping and changes in BMI. With regard to our sensitivity analysis, no association between breakfast skipping and overweight/obesity or BMI change, respectively, was found.

The results of this report are similar to recent systematic reviews and meta-analyses on studies with cross-sectional designs, but with a lower strength of association observed in our results. For comparison, the relative risk for overweight/obesity was increased by about 75% (95% CI: 57%, 95%) for the breakfast skippers compared to the breakfast eaters analyzed in the meta-analysis on cross-sectional studies from Asian and Pacific countries [95]. In contrast, the results of meta-analyses on studies with interventional designs reported associations between breakfast skipping and body weight concerning weight loss in breakfast skippers [63,96]. In accordance, the recent meta-analysis by Bonnet et al. [63] reported a weighted mean difference of −0.54 kg (95% CI: −1.05 kg, −0.03 kg) in body weight when breakfast was skipped in trials conducted in the UK and USA, with a follow-up time between 4 and 16 weeks.

In addition, breakfast skipping is part of several intermittent fasting programs [97,98,99]. One intermittent fasting method is time-restricted feeding, whereby the individual fasts for 16 to 20 h per day and eats only in the remaining 4 to 8 h per day, mainly in the evening. This fasting program is called “20:4” or “16:8”, respectively [97,98,100,101]. Systematic reviews on intermittent fasting programs [97,98] suggest that body weight reduction is possible. This association is stronger in interventional and randomized controlled studies than in observational studies [97,98]. Those results stand against the findings of the present review and meta-analyses and might be due to the different study designs and outcome measurements.

Randomized controlled trials are considered the gold standard study design, they are conducted under laboratory conditions, provide good internal validity, and are labeled with the highest level of evidence. Typically, their outcome measurements are more trustworthy than outcome measurements in observational study designs. In view of overweight and obesity, trials utilized body composition values (e.g., fat mass and fat free mass) by using DEXA or BIA, respectively [63,96,102], whereas observational longitudinal studies utilized BMI or weight change in kilograms by using a tapeline and scale [36,37,38,39,90,91,92,93,94]. However, trials such as those conducted by Sievert et al. [96] and Bonnet et al. [63] are limited to a small sample size (i.e., <500 people included in meta-analysis) and short observation intervals (i.e., between 2 and 16 weeks). In contrast, our meta-analyses on cohort studies contained ≥ 100,000 participants. Furthermore, those observational longitudinal studies followed-up the study populations after between 5 and 18 years [36,37,38,39,90,91,92,93,94]. The main advantage of observational longitudinal studies is that they are able to determine real-world conditions. Their results provide better external validity and are more transferable to the general population than the results of trials. A final difference to take into account is the fact that the analyzed population in five out of seven trials included in the meta-analysis by Bonnet et al. [63] (or five out of ten trials included in the study by Sievert et al. [96]) were overweight/obese, and the population was younger, with a mean age of 35 years.

The physiological principles of intermittent fasting are interesting in view of weight changes. The metabolic conversion of receiving energy from the glycogen stores of the body starts 12 h after the last ingestion of food. A few days later, up to 90% of energy supply stems from the adipose tissue. This has the clinical effect of losing the visceral fat. With losing this fat, the body weight and the metabolic health risk decreases, but levels of the hormone leptin increase. The latter causes a ravenous appetite [99]. People who skipped breakfast are more likely to crave for high caloric food than low caloric food [103]. In accordance with this, the issues of whether breakfast’s satiating effect takes influence on the TEI and whether breakfast eating or skipping increases the TEI have been discussed. Some studies have reported a lower TEI in breakfast eaters compared to skippers [104,105,106], while other studies reported a higher TEI [41,71,75,96,107].

Breakfast is only one of several determinants on body weight, and even in the scope of breakfast itself, there are different factors influencing the body weight status through nutritional physiological processes. For instance, the time of the day at which breakfast is eaten (e.g., before 10 a.m.), time spent on eating (e.g., at least 20 min), its energy intake (e.g., containing 25% of TEI) and its composition (e.g., wholegrain-based, fiber-rich foods) are associated with the body’s weight status due to the release of gut hormones [62]. For example, the analysis by Deshmukh-Taskar et al. [40] suggests that ready-to-eat cereals are the best kind of breakfast for losing and maintaining body weight. This difference between type of breakfast and body weight status has also been seen in other studies [41,108]. Additionally, the analysis by Kent et al. [109] shows that the larger the breakfast proportion size, the lower the BMI of men. The observed relationship was even more pronounced in vegetarian men, compared to their non-vegetarian counterparts [109]. This opens a new viewpoint which should be considered when looking at the relationship between breakfast skipping and body weight.

Since the nourishment of European, American, and Asian breakfasts are not comparable [36,92,94], one needs to consider that the current meta-analyses pooled breakfasts from countries with different breakfast types. Therefore, this analysis is limited to the extent that the examined results are most likely not transferable to the context of an individual country.

Furthermore, breakfast is an indicator of general health-promoting lifestyle and behavior: breakfast skippers are more likely to be smokers compared to breakfast eaters [34,40,74,76,77,78,80,110]. With a decreasing number of days on which breakfast is consumed per week the likelihood to be a smoker increases [76]. Likewise, breakfast skipping is associated with a higher level of alcohol consumption [34,40,74,110]. Breakfast skippers drank on average 20.5 g alcohol per day, whereas breakfast eaters drank averagely 11.9 or 8.6 g alcohol per day, respectively. Ready-to-eat cereal breakfast consumers drank less alcohol than consumers of other types of breakfast [110]. As well, breakfast skippers are more likely to be physically inactive than breakfast eaters [34,70,71,74,76,77,78,110]. Moreover, breakfast skipping is correlated to a worse quality of sleep [76,77,78]. Furthermore, breakfast skippers are more likely to have deficits in macro- and micronutrient intake [41,71,106,107,110,111]. In addition, breakfast skippers consumed the highest content of added and free sugar [71]. Lastly, breakfast skippers are more likely to have lower scores of general health perceptions, vitality, social functioning, emotional roles, mental health, and total health status scores compared to breakfast eaters [74]. Therefore, these factors should be adjusted, as was done in some of the included studies.

Moreover, there are socioeconomic and demographic differences in characteristics of breakfast skippers compared to breakfast eaters [34,40,71,76,77,79,80,110,112]. People affected by poverty are most likely to skip breakfast [40,110], whereas people affiliated by the highest socioeconomic status are most likely to consume cereals for breakfast [71,79]. As well, being married seems to increase the likelihood of breakfast consumption [76,79,110]. Breakfast skippers tend to be of younger age, so the likelihood of breakfast consumption increased with increasing age [34,40,71,79,80,113]. Likewise, the literature shows a sex gradient: men are more likely to skip breakfast than women [34,40,112]. Lastly, there are differences between ethnicities displayed in the literature. Breakfast skippers are more likely to be of non-Hispanic black ethnicity and are less likely to be of non-Hispanic white ethnicity than breakfast eaters [75,110].

With regard to the behavioral, demographic and socioeconomic factors which are associated with breakfast skipping, most of them are also associated with overweight/obesity and may, therefore, lead to confounding factors: people with overweight/obesity are more often physically inactive compared to individuals of healthy weight [73,114,115,116]. An unhealthy diet is also seen more often in people with overweight/obesity than in people with normal weight [68,73]. Additionally, BMI decreased in people who smoked compared to non-smokers [68]. Overweight/obese people are more likely to have a lower socioeconomic status and/or have a lower educational level [72,117]. Another study [116] reported that BMI increased with increasing economic status in both women and men. With increasing age, the likelihood of overweight/obesity also increased [67]. The sex gradient indicates that women are more likely to be overweight/obese, globally [118]. Moreover, the prevalence of overweight/obesity is positively linearly related to the income level of a country. The higher a country’s income level, the higher the prevalence [118]. Besides income-dependent differences in the prevalence of overweight/obesity, a variety in fat distribution regarding ethnicity could have an influence [9].

### 4.1. Limitations

The described issue of an insufficient number of studies included in the meta-analyses, is one of the main limitations of this study. This is likely leading to a publication bias and a small study effect but it was not practicable to check for this with only two included studies for each meta-analysis. With regard to this, an overestimation of the true effect size is likely [83]. Due to this, this study is not able to estimate the range in which the true effect is located with an appropriate level of precision. This is also visible when looking at the reported 95% CIs, which are relatively broad. Additionally of note, a second limitation is based on the fact that studies included in the analysis show some differences in their methodologies. For example, we pooled data of a primary and secondary nature and with different outcome measurements (i.e., BMI change/year and BMI change without a clear period). Moreover, the sensitivity analyses (standard error adjustment to face the heterogeneity) displayed no association between breakfast skipping and overweight/obesity or BMI change, respectively. Accordingly, the reported results are not robust for variations [83]. Additionally, information on differences in this association dependent on the composition, quantity, and quality of breakfast cannot be displayed. Brikou et al. [119] report that the definitions of breakfast and the classifications of being a breakfast skipper/eater vary highly. As most studies did not report a definition of breakfast or used different definitions of breakfast, a misclassification bias must be considered. Stemming from the fact that this meta-analysis did not comprise a sufficient number of studies to stratify for the quality or measurement methods of the studies, we were not able to provide suggestions on the influence on the analyzed association.

In the scope of significance testing, it is also worthy of note that the larger the analyzed sample size, the higher the likelihood for a statistically significant result (and rejecting H0). The present meta-analyses contained a large sample (approx. 100,000) but very small number of included studies (i.e., two) and showed small effect sizes (i.e., RR = 1.11 and β = −0.02). Therefore, a false negative effect (i.e., beta error) should be taken into account, as well [120].

Furthermore, the study with the comparably highest risk of bias (i.e., critical risk) [37] received the largest weighting (i.e., 75%) within the first meta-analysis. The risk of bias in the other two studies included in the meta-analyses also rated high (i.e., serious risk). Additionally, the provided meta-evidence is of a very low level for both meta-analyses.

The measurement methods of breakfast skipping behavior were different between the included studies. Only one out of nine studies used an interview-based assessment, while the residual eight studies used a questionnaire-based assessment of breakfast skipping. This should be taken into account, because the meta-analysis by Horikawa et al. [95] implied that interview-based assessment of breakfast skipping is more strongly associated with overweight/obesity than the questionnaire-based assessment of breakfast skipping. Therefore, it could be suggested that the results of this thesis are limited in their precision. Another limiting factor is confounding in the included studies. Only two out of nine studies adjusted for all seven important variables [90,91]. Because most studies did not adjust their analyses for these important covariables, confounding due to missing adjustment, and a residual confounding must be considered. Finally, this systematic review has not been registered prior to execution.

### 4.2. Strengths and Further Research Needs

To the authors’ best knowledge, this is the first meta-analysis on the association between breakfast skipping and body weight, examined in the form of an observational longitudinal study of adults, worldwide. Furthermore, the results of these meta-analyses are strengthened due to the observational longitudinal nature of the included cohort studies. They represent real-world conditions of the association between breakfast skipping and body weight. Cohort studies avoid recall bias and decrease the potential for selection bias [120].

Considering the reported results of previous and current research, there are still open questions left and new questions accrued which collectively implies a further need for research. In general, there is a need for a clear and consistent definition of breakfast eating and breakfast skipping. Breakfast behavior is mostly not part of a validated measurement tool when looking at the included studies of this analysis. It could be an aim in future research to develop a validated measurement tool for breakfast eating and breakfast skipping.

Moreover, this review displayed a lack of studies conducted in European and African countries compared to the number of studies from the USA and Japan. In addition, future observational studies should consider other measurement methods for body weight/body composition than only BMI. Randomized controlled trials already used these specific outcome measurement methods but analyzed only short-term changes in body weight. Large observational longitudinal studies equipped with those outcome measurement methods would hold an advantage to answer the question of whether breakfast is a relevant setscrew aspect of lifestyle to consider in body weight management to avoid overweight-related deuteropathies. Comparatively, long-term randomized trials would be helpful to examine the association itself with a great deal of evidence but with less transferability into lifestyle. Lastly, both designs—trials and cohorts—are important in examining the association between exposure and outcome at first and working out this association under real and heterogenic life conditions subsequently. For another comparison, there is a need for studies with observational longitudinal design on the association between intermittent fasting and body weight, to compare those results with the results of breakfast skipping/eating and body weight.

## 5. Conclusions

In conclusion, evidence from observational studies indicates that in real-world settings, breakfast skipping might lead to weight gain and the onset of overweight and obesity. However, the findings rely on very few studies with high levels of heterogeneity and therefore these need to be interpreted with caution. Further observational longitudinal studies on this topic which use a clear and consistent definition of breakfast eating and breakfast skipping are needed. In addition, future studies should focus on further anthropometric measures besides BMI and consider potential confounders.

## Figures and Tables

**Figure 1 nutrients-13-00272-f001:**
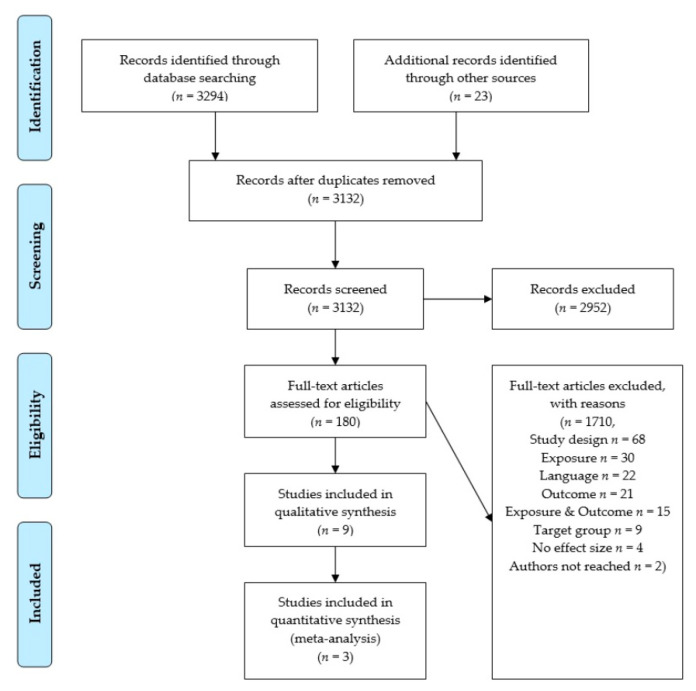
Preferred Reporting Items for Systematic Reviews and Meta-Analyses (PRISMA) flowchart [89] of the literature screening on the association between breakfast skipping and body weight.

**Figure 2 nutrients-13-00272-f002:**
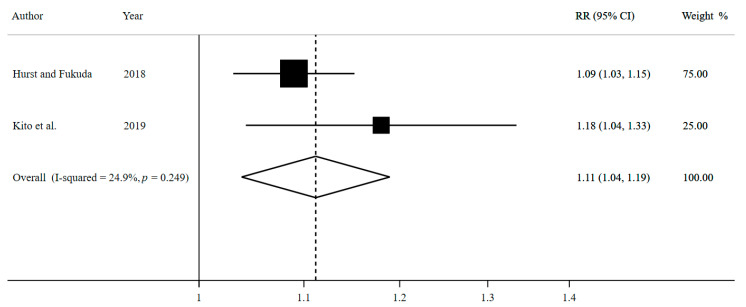
Forest plot for breakfast skipping ≥3 versus ≤2 days per week and overweight/obesity by using random-effects meta-analysis.

**Figure 3 nutrients-13-00272-f003:**
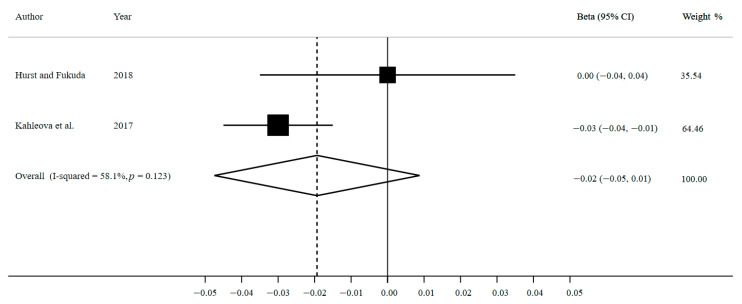
Forest plot for breakfast eating versus breakfast skipping and BMI change by using random-effects meta-analysis.

**Table 1 nutrients-13-00272-t001:** Characteristics of the included studies.

Reference	Country	Study Design and Follow-Up Period	Participants		Exposure vs. Comparison, Measurement	Outcome, Measurement	Results
			*n* Total *n* Cases *	Sex ** Age ***			Estimated Effect Sizes (95%CI LL; UL)
Goto et al., 2008 [36]	Japan	retrospective, check-up data, 2000–2007	4634 598	100% 21.5	Skipping ≥ 2 vs. ≤ 1 d/wk, self-administered questionnaire	>5% increased BMI, weight and height measurement	OR = 1.34 (1.12; 1.62)
Guinter et al., 2020 [90]	USA, Puerto Rico	Sisters Study, prospective cohort, 2003–2015	46,037 2797, 2383, 6807	0% 55.3	Eating 3–4 d/wk vs. 0, 1–2, 5–6, 7 d/wk, FFQ	5-year incident BMI ≥ 25 kg/^2^, ≥ 30 kg/^2^, ≥ 5 kg weight gain, weight and height measurement and self-reported weight	5-yr incident BMI ≥ 25kg/^2^: 0 d/wk RR = 0.74 (0.62; 0.89), 1–2 d/wk RR = 0.91 (0.78; 1.07), 5–6 d/wk RR = 0.97 (0.85; 1.09), 7 d/wk RR = 0.88 (0.78; 0.99) 5-yr incident BMI ≥ 30kg/^2^: 0 d/wk RR = 0.72 (0.59; 0.87), 1–2 d/wk RR = 0.75 (0.62; 0.89), 5–6 d/wk RR = 0.91 (0.80; 1.04), 7 d/wk RR = 0.79 (0.70; 0.90) 5-yr incident ≥ 5 kg weight gain: 0 d/wk RR = 1.00 (0.90; 1.11), 1–2 d/wk RR = 0.98 (0.89; 1.08), 5–6 d/wk RR = 0.99 (0.92; 1.06), 7 d/wk RR = 0.97 (0.91; 1.04)
Hurst and Fukuda, 2018 [37]	Japan	secondary analysis of insurance and health check-up data, 2008–2013	59,717 20,671	66% 47.4	Skipping ≤ 2 vs. ≥ 3 d/wk, Health check-up question	BMI ≥ 25 kg/^2^, BMI and WC change, BMI and WC data from check-up	BMI ≥ 25kg/^2^: OR = 0.92 (0.87; 0.97) BMI change (in kg/^2^): β = 0.00 (−0.03; 0.04) WC change (in cm): β = 0.03 (−0.11; 0.16)
Kahleova et al., 2017 [93]	North America Canada	AHS-2, prospective cohort, 2002–2010	50,660 n.g. ^++^	36% 58	Eating vs. skipping, Hospital History Form	BMI change/year, weight and height measurement and self-report	BMI change (in kg/^2^): β = −0.03 (−0.04; -0.01)
Kito et al., 2019 [38]	Japan	retrospective cohort, 2008/09–2012	45,524 5093	100% 34	Skipping ≥ 3 vs. ≤ 2 d/wk, Health check-up question	BMI ≥ 25 kg/^2^, weight and height measurement	OR = 1.18 (1.04; 1.33)
Nooyens et al., 2005 [92]	The Netherlands	Doetinchem Cohort Study, prospective, 1987–2002	288 n.g. ^++^	100% 54.9	Eating 0–7 d/wk Dutch version of EPIC FFQ	Weight and WC change/year, weight, height and WC measurement	Weight change (in kg): β = 0.04 (n.g. ^++^) WC change (in cm): β = 0.10 (n.g. ^++^)
Odegaard et al., 2013 [91]	USA	CARDIA Study, prospective cohort, 1992/93–2011	3598 972 WC 783 BMI	44% 32.1	Eating ≤ 3 vs. 4–6, 7 d/wk, interviewer-administered CARDIA DHQ	BMI ≥ 30 kg/^2^, WC > 88 cm for women and > 102 cm for men, weight, height and WC measurement	BMI ≥ 30 kg/^2^: 4–6 d/wk HR = 0.75 (0.62; 0.90), 7 d/wk HR = 0.57 (0.47; 0.68) WC > 88 or 120 cm: 4–6 d/wk HR = 0.84 (0.70; 0.99), 7 d/wk HR = 0.78 (0.66; 0.91)
Smith et al., 2017 [39]	Australia	CDAH Study, prospective cohort, baseline 2002/04–2011	1155 410	43% 31.5	Met guidelines ^#^ consistently vs. met not, postal questionnaire	5-year weight change, weight and height measurement and self-report	5-yr weight change (in kg): β = 1.5 (0.5; 2.8)
van der Heijden et al., 2007 [94]	USA	HPFS, prospective cohort, 1992–2002	20,064 5857	100% 57.3	Eating vs. skipping, semi-quantitative FFQ	≥ 5 kg weight gain, self-reported weight	HR = 0.87 (0.82; 0.93)

* participants, in which the analyzed outcome occurred; ** male sex in percentage; *** mean age in years; 95% CI LL; UL = 95% confidence interval: lower limit; upper limit; d/wk = days per week; ^++^ information not given; AHS = Adventist Health Study; CARDIA = Coronary Artery Risk Development in Young Adults; CDAH = Childhood Determinants of Adult Health; HPFS = Health Professionals Follow-Up Study; FFQ = Food Frequency Questionnaire; EPIC = European Prospective Investigation into Cancer and Nutrition; BMI = body mass index; WC = waist circumference, kg = kilograms; m^2^ = square meters; yr = year; OR = odds ratio; RR = relative risk; HR = hazard ratio; ^#^ referring to the 2010 Dietary Guidelines for Americans: eating a nutrient-dent breakfast vs. not eating breakfast [35].

**Table 2 nutrients-13-00272-t002:** Adjusted confounders in included studies.

Reference	Adjustment for Important Variables	All 7	Adjustment for Other Variables
Age	Sex	Education	Smoking	Physical Activity	Alcohol	TEI
Goto et al., 2008 [36]									fatty food, living alone
Guinter et al., 2020 [90]									race/ethnicity, Healthy Eating Index 2015, weight loss dieting, average sleep hours, perceived level of stress
Hurst and Fukuda, 2018 [37]									baseline BMI, obesity status, antidiabetic medication
Kahleova et al., 2017 [93]									ethnicity, dietary pattern, marital status, sleep, tv watching, high blood pressure medication
Kito et al., 2019 [38]									BMI, eating speed, late-night meals/ snacking, drinking, sleep, interactions
Nooyens et al., 2005 [92]									retirement, type of job, diet, sugared soft drinks, fiber density, interactions
Odegaard et al., 2013 [91]									race, fast food, dietary quality, meal frequency, baseline BMI and WC
Smith et al., 2017 [39]									baseline weight, time to follow-up, meal pattern, weekday of follow-up
van der Heijden et al., 2007 [94]									baseline BMI, marital status, weightlifting

TEI = total energy intake; BMI = body mass index; WC = waist circumference.

## Data Availability

Not applicable, because it is a systematic review with meta-analyses. All data is available in primary studies.

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
