# Peer review of "Association between Breakfast Skipping and Body Weight—A Systematic Review and Meta-Analysis of Observational Longitudinal Studies"

_nutrients, 2021, doi:10.3390/nu13010272_

Round 1

Reviewer 1 Report

Dear Authors

The study is giving new evidence on the relationship between overweigh/obesity and skipping breakfast,  the study focused on adults; it would be goof if you include school children age groups.

However the study is excellent although the sample size is small. 

I recommend to expand the introduction and include more data on obesity and its relation to NCDs as a burden, and link it with the WHO recommendations.

You may need to add few recommendations to the public, as an outcome to your study.   

Author Response

Please find a point-by-point response in the document attached.

Reviewer 2 Report

Dear Authors,

Breakfast skipping and body weight is an important topic and this review and meta-analysis provide an overview of the current state of knowledge.

The following minor revisions are recommended for improvement before publication.

Introduction

line 33: The link for reference is not working. Please check if more recent data are available (World Obesity, IOTF wbsite?)

lines 53-61: In this section I would like to have a little more information on the relationship between breakfast and body weight: potential mechanism, hypotheses, etc.

Materials and Methods

Why the following criteria for the frequency of eating breakfast were adopted: ≥ 3 days and ≤ 2 days? Information is lacking on this, please explain.

Results

This section is well described, however, analyzes were conducted on the basis of a few studies only and with some differences in methodology. It is not the authors' fault, but it should be emphasized in the discussion as a certain limitation of the work. 

Discussion

General: Are the results obtained by the authors consistent with the results of similar analyzes but concerning the pediatric population?

Author Response

(The authors gave the same response as above.)
